# Flexible Syndesmotic Reconstruction with Two Suture Buttons Provides Equal Stability Compared to Syndesmotic Screws: A Biomechanical Study

**DOI:** 10.3390/bioengineering12070685

**Published:** 2025-06-23

**Authors:** Alexander Milstrey, Vivienne Hoell, Ann-Sophie C. Weigel, Jens Wermers, Stella Gartung, Julia Evers, Michael J. Raschke, Sabine Ochman

**Affiliations:** 1Department of Trauma-, Hand- and Reconstructive Surgery, University Hospital Muenster, 48149 Muenster, Germany; 2Department of Trauma Surgery and Orthopedics, Protestant Hospital of the Bethel Foundation, Bielefeld University, 33615 Bielefeld, Germany

**Keywords:** syndesmotic instability, syndesmotic screw, Tightrope, Suture Button, biomechanic

## Abstract

*Background*: This study investigated syndesmotic stability after transection and the effects of stabilization using rigid and dynamic reconstruction techniques. *Methods*: Syndesmotic stability was analyzed using a six-degree-of-freedom robotic arm on 14 human specimens. Stability was analyzed in the neutral position and during dorsiflexion and plantar flexion using an external rotation stress test under an axial load of 200 Newtons. The examination was performed on intact and sequentially transected syndesmosis in the following order: (1) anterior inferior tibiofibular ligament (AITFL); (2) interosseous ligament (IOL); and (3) posterior inferior tibiofibular ligament (PITFL). Then, reconstruction was performed using different syndesmotic screw techniques or a dynamic Suture Button system (Arthrex TightRope; n = 7). *Results*: A syndesmotic transection mainly caused sagittal instability of the fibula. While both static and dynamic reconstruction provided stabilization, screw fixation, particularly with two screws and a plate, demonstrated superior control of the fibular movement, especially in the sagittal and transverse planes, compared to one Suture Button. *Conclusions*: The results suggest that syndesmotic stabilization with one Suture Button may be insufficient for cases involving three-ligamentous injuries, whereas two Suture Buttons may offer comparable biomechanical stability to syndesmotic screws. Additionally, the study suggests that lateral radiographs may provide additional clinical value in assessing syndesmotic stability.

## 1. Introduction

Injuries to the syndesmosis, which are often the result of high-energy trauma, can lead to significant ankle instability and long-term complications. Epidemiological data indicate that syndesmotic injuries are underdiagnosed, with a prevalence ranging from 1% to 18% of all ankle injuries [1,2]. The most commonly investigated mechanism of injury involves a combination of dorsiflexion, external rotation, and eversion of the foot because this movement stresses the syndesmotic ligaments [3,4].

The goal of surgical intervention is to restore anatomical kinematics in the distal tibiofibular joint. Traditionally, syndesmotic screws have been used for this purpose; however, alternative fixation methods, such as Suture Button systems, have emerged over the past decade [5,6,7,8].

Clinical studies, including randomized controlled trials (RCTs), have also investigated the differences between syndesmotic screw and Suture Button fixation. Some RCTs report equivalent results between Suture Button systems and screw fixation, showing similar functional outcomes and patient satisfaction [5,6]. In contrast, other studies have suggested that Suture Button systems may offer advantages, such as earlier weight-bearing and a reduced need for subsequent hardware removal [7]. Dynamic systems allow micromovements that correspond to the physiological state, potentially resulting in better regeneration of the syndesmosis and a reduced rate of misalignment of the fibula into the incision [9]. However, recent biomechanical studies show that one dynamic system alone does not lead to sufficient tibiofibular stability, particularly in external rotation and sagittal translation. Additional procedures are necessary [10].

However, clinical findings often focus on functional outcomes and complications, with less emphasis on the detailed biomechanical characteristics of the distal tibiofibular joint.

This study aims to compare the biomechanical efficacy of different screw and Suture Button fixation methods in restoring syndesmotic stability. Our hypothesis is that sequential cutting of the distal tibiofibular joint increases syndesmotic instability proportionally and that the screw fixation methods provide equal stability to the two Suture Button fixation methods.

## 2. Materials and Methods

### 2.1. Specimen Preparation

Fourteen unpaired, fresh-frozen human cadaveric lower leg specimens (mean age 81.8 years; range 62–93 years) with no history of previous injury or rigid deformity were used. Radiography was performed to exclude specimens with evidence of osseous or of degenerative abnormalities. The specimens were thawed 24 h before testing, and the soft tissues were carefully removed while preserving the ligamentous structures. Two angular-stable proximal tibial plates and three cancellous spanning subtalar screws were implanted. The calcaneus was embedded in a metal alloy (Cerrobend, Bolton Metal Products Co., Bellafonte, PA, USA) and attached to the robotic arm, as described previously [11].

### 2.2. Biomechanical Setup

A 6-axis industrial robot (KR 60/3, KUKA, Augsburg, Germany) with an accuracy of ±0.06 millimeter (mm) was used to apply controlled motions and loads to the ankle specimens. The kinematics of the ankle were measured with a six-degree-of-freedom force/torque load cell with an accuracy of ±0.25 Newton (N) and ±0.05 Newtonmeter (Nm). An optical tracking system (Optotrak Certus Motion Capture System; Northern Digital, Waterloo, ON, Canada) recorded the three-dimensional movement of the tibia and fibula. A custom software platform, incorporating an ankle tool for robotic simulation (SimVitro, Cleveland Clinic BioRobotics Lab, Cleveland, OH, USA), was employed. System optimization for subsequent movements involved performing a passive path from 10° dorsiflexion to 20° plantarflexion. This movement was achieved with motion-controlled flexion, 50 N compression, and constrained forces in all other directions. The coordinate system was established with the x-axis oriented laterally, defined by a line from the medial to the lateral malleolus at the joint line. The z-axis was set perpendicular to the x-axis, residing within the plane of the tibial shaft axis. The y-axis was positioned posteriorly and was perpendicular to the other two axes. These axes formed the basis for defining the coronal, sagittal, and axial planes. The origin of this coordinate system was precisely located at the lateral border of the distal fibula, at the ankle joint level. Coronal and sagittal displacements are presented as relative lateral and dorsal shifts of the fibula, measured in millimeters (mm). Axial motion is quantified as relative external rotation in degrees (°), with all translations referenced to the specific neutral position established at the initiation of each step in the study protocol.

### 2.3. Study Protocol

The study protocol consisted of two parts. First, the syndesmotic region of all specimens was cut in a sequential cutting order (steps 1–3). Afterwards, the specimens were randomized to either different syndesmotic screw reconstructions (steps 4–7) or two different Suture Button methods (steps 4–5, see Figure 1).

With a continuous axial compression of 200 N applied to simulate partial weight bearing, the specimens were moved from 10° dorsiflexion to 20° plantarflexion, covering a physiological range of motion. The ankle’s three-dimensional response to an externally applied 5 Nm external rotation torque was quantified. This measurement was taken at 10° dorsiflexion, in the neutral position and at 20° plantarflexion.

### 2.4. Sequential Cutting

The syndesmotic ligaments were sectioned in the following order: anterior inferior tibiofibular ligament (AITFL), interosseous ligament (IOL) including the distal 5 cm of the interosseous membrane, and posterior inferior tibiofibular ligament (PITFL) (steps 1–3). Measurements were taken after each transection.

### 2.5. Syndesmotic Screw Fixation

Prior to the syndesmotic transection, two 2.5 mm holes were predrilled two and three centimeters above the ankle joint, running parallel to the joint line. The syndesmosis was reconstructed in the following sequence: (1) one tricortical 3.5 mm titanium screw; (2) two tricortical 3.5 mm screws; (3) two tricortical screws with a two-hole, one-third titanium tubular plate; and (4) two quadrocortical 3.5 mm screws (all screws/plates: DePuy Synthes, Raynham, MA, USA) (steps 4–7).

### 2.6. Suture Button Fixation

Prior to the syndesmotic transection, the Suture Buttons (TightRopeTM, Arthrex, Naples, FL, USA) were predrilled two and three centimeters above the ankle joint and parallel to the joint line in the coronal plane. The syndesmosis was reconstructed with one Tightrope^TM^ two centimeters above the joint line and directed anteriorly to mimic the reconstruction of the AITFL. Next, a second divergent Tightrope was implanted one centimeter above the first one and directed posteriorly to mimic a PITFL reconstruction (steps 4–5).

### 2.7. Statistical Analysis

An A-priori power analysis was performed using G*Power (version 3.1.9). Based on means and standard deviations from previous biomechanical studies testing syndesmotic instability [12], it was determined that a sample size of seven per group would be able to detect changes in rotation of 1.0° (with a standard deviation of 0.8°) and translation of 1.0 mm (with a standard deviation of 0.8 mm) with 95% power at the *p* < 0.05 significance level. The results are presented as the means of each group with standard deviation. Figures are presented in box plots, with the median of each group in the center line, the box drawn from the first to the third quartile, and whiskers representing the minimum and maximum of each group. All data were tested for normal distribution with a Shapiro–Wilk test. Since the cutting sequence included 14 specimens and the reconstruction sequence included only 7 specimens per group, all data were analyzed using a mixed-effects analysis with a Tukey’s multiple comparisons test. In all cases, significance was set at *p* < 0.05. Fibular instability was defined as a significant increase in motion in response to the external rotation stress test in at least one direction. All significant differences are shown with horizontal lines in Figures 2–4. Prism 10 software was used for statistical tests (GraphPad Software Inc., La Jolla, CA, USA).

## 3. Results

### 3.1. Cutting of the Syndesmosis

Sequential syndesmotic transection primarily resulted in increased posterior translation of the fibula. The greatest increase in posterior translation occurred after cutting the AITFL. The posterior displacement of the fibula was 7.2 mm ± 1.3 mm in the intact state, 10.6 mm ± 2.6 mm with AITFL cutting (*p* < 0.001), 11.0 mm ± 3.0 mm with IOL cutting (*p* = 0.001), and 12.7 mm ± 2.8 mm (*p* < 0.0001) with PITFL cutting in a neutral ankle position.

Sectioning the AITFL did not result in significant coronal translation of the fibula. The medial movement of the fibula only became significant after cutting the interosseous ligaments or completely disrupting the syndesmosis (intact: 0.71 mm ± 1.1 mm; IOL: 1.3 mm ± 1.1 mm; PITFL: 2.1 mm ± 1.4 mm). 1.0 mm ± 1.0 mm; IOL: 1.3 mm ± 1.1 mm; PITFL: 2.1 mm ± 1.4 mm).

External rotation increased in all cutting sequences compared to the intact state, except for the PITFL at 20° of plantarflexion. The main increase occurred with transection of the AITFL. This resulted in external rotation of the fibula of 5.0° ± 1.3° in the intact state; 6.9° ± 1.8° with AITFL transection (*p* < 0.001); 6.2° ± 1.7° with IOL transection (*p* = 0.02); and 6.7° ± 1.8° (*p* = 0.004) with PITFL transection in a neutral ankle position.

### 3.2. Syndesmotic Reconstruction

In neutral position, there was a significant increase in posterior translation with screw fixation using one tricortical screw (*p* = 0.04), two tricortical screws (*p* = 0.01), or two quadrocortical screws (*p* = 0.02), as well as with fixation using one Tightrope (*p* = 0.03). However, two tricortical screws and a plate (*p* = 0.43) and two Suture buttons (*p* = 0.23) were the only constructs that restored the fibular sagittal kinematics compared to the intact state (see Figure 2a). In the coronal plane, all reconstruction techniques, including screw and Suture button fixation, effectively restored fibular stability. In the transverse plane, two tricortical screws, but not a single screw, reduced external rotation of the fibula compared to the injured syndesmosis (*p* = 0.01). Suture button constructs did not limit external rotation (see Figure 2).

With the ankle in 10° dorsiflexion, both the single screw construct (*p* = 0.05) and the two-screw with plate construct (*p* = 0.02) significantly reduced posterior fibular translation. Remaining sagittal instability occurred with two quadrocortical screws (*p* = 0.04) and one Suture Button (*p* = 0.04). In the coronal plane, all reconstruction techniques restored fibular stability. Screw fixation demonstrated better control of fibular rotation in the transverse plane than Suture Button systems, with a significant decrease in rotation compared to the injured syndesmosis. This decrease was only achieved with two tricortical screws with a plate (*p* = 0.03), and there was only a remaining instability with one Suture Button (*p* = 0.02) compared to the intact state (see Figure 3).

At 20° of plantarflexion, all screw constructs effectively reduced posterior fibular translation. In the coronal plane, all reconstruction techniques restored fibular stability. Screw fixation provided equivalent control of fibular external rotation in the transverse plane when compared to Suture Button systems with a remaining instability compared to the intact state only with one tricortical screw (*p* = 0.001) compared to the intact state (see Figure 4).

## 4. Discussion

This study confirms that syndesmotic injury leads to increased fibular movement in all three planes. The results suggest that screw fixation, particularly with two screws or a screw-plate construct, provides superior biomechanical stabilization, especially in controlling posterior translation and external rotation of the fibula. However, syndesmotic stabilization with one Suture Button may be insufficient for three-ligamentous unstable syndesmotic injuries because it fails to provide adequate control in the sagittal and transverse planes. However, the use of two Suture Buttons appears to offer stability similar to that achieved with screw fixation.

A significant finding of this study is the prevalence of fibular instability, primarily in the sagittal plane, following syndesmotic injury. The fibula demonstrated increased posterior translation after ligamentous injury compared to lateral instability (5.5 mm vs. 1.2 mm in the neutral position). This finding is supported by other biomechanical research [11,13]. Candal-Cauto et al. observed a significant dorsal shift of 8.8 mm compared to only 1.5 mm of lateral fibular translation [13]. Therefore, a hook test or external rotation stress test, which assess posterior fibular translation on a lateral radiograph, may be more effective at detecting syndesmotic instability than the standard mortise view.

Since even a 1 mm shift in the fibula can reduce the tibiotalar contact area by up to 42%, syndesmotic instabilities are commonly treated surgically [14]. Although Suture Button fixation has become more popular, syndesmotic screws have long been the standard surgical method for stabilizing unstable syndesmotic injuries. However, debates regarding the optimal number and length of screws, as well as the potential benefits of adding a plate, are ongoing. Few biomechanical studies have evaluated the optimal syndesmotic screw technique [15,16,17,18,19,20]. A larger diameter (4.5 mm vs. 3.5 mm) does not seem to provide greater biomechanical stability [18]. However, smaller 3.5 mm screws break more often than larger 4.5 mm screws, without resulting in significant functional differences [21,22]. Regarding screw length, both biomechanical and clinical studies have shown no significant differences between tricortical and quadrocortical screws [17,18]. Although biomechanical studies have shown that two syndesmotic screws provide more stability, there is currently no clinical evidence that a higher screw count improves functional outcomes [16,23,24]. Only one biomechanical study evaluated a two-hole plate with two 3.2 mm locking screws, which resulted in equal stability during axial loading, but increased the external rotational load to failure compared to a two-hole locking plate with two 4.5 mm quadrocortical screws [25]. The results of our study are consistent with the existing literature, with only minor differences between the screw designs. This study is the first to compare these techniques at different degrees of dorsiflexion and plantarflexion, as well as to compare them with two flexible reconstruction techniques. While both screw and Suture Button fixation methods can restore stability in the coronal plane, screw fixation—especially with two screws or a screw-plate construct—seems to provide better control of fibular movement in the sagittal and transverse planes.

In more severe or complex unstable syndesmotic injuries (e.g., Maisonneuve fractures), a second Tightrope is often used to improve the stability of the repair. To further improve rotational stability, the second Tightrope is typically inserted at a slightly different angle than the first. Currently, there are no universally accepted guidelines that definitively indicate when a second Tightrope is necessary. However, Kurtoglu et al. reported no statistically significant difference in functional outcomes in a retrospective case series comparing 23 patients treated with one Suture Button and 20 patients treated with two [26]. Furthermore, previous biomechanical studies have also shown no significant difference with one or two Tightropes [27,28,29]. Tsai et al. showed an equal maximum external rotational force in a load to failure study comparing one to two Tightropes [28]. Parker et al. demonstrated an equal biomechanical stability of one Tightrope, two parallel Tightropes and two divergent Tightropes in response to a 7.5 Nm external rotational torque in a neutral ankle position [29]. This suggests that a single Tightrope is sufficient to achieve satisfactory functional results for many unstable syndesmotic injuries. However, our study indicates that stabilization with two Tightropes restores syndesmotic kinematics equally well as different syndesmotic screw techniques. A single Tightrope, on the other hand, may be more unstable. Our model is more precise than previous literature in detecting differences between the two Tightrope techniques because we tested the kinematics at different degrees of dorsiflexion and plantarflexion of the ankle. The placement of subtalar screws provided optimal control of talar kinematics, increased the rigidity of hindfoot fixation, and allowed the robotic arm to directly control the hindfoot.

Therefore, to the best of our knowledge, this is the first study to confirm the theory of enhanced biomechanical stability in the sagittal and axial planes when the syndesmosis is reconstructed with two Tightropes rather than one. However, the significant findings of our study may not be clinically relevant, particularly since there is no consensus on what constitutes a clinically relevant level of rotational instability. Therefore, future clinical studies should focus on establishing a relevant threshold for syndesmotic instability in all three planes.

This study has several limitations. First, the use of cadaveric specimens with a median age of 81 years may not fully represent the biomechanical properties of younger, active individuals typically affected by syndesmotic injuries. The absence of musculature in the cadaveric model limits the ability to replicate dynamic joint stability. Similar to clinical application, the Suture Buttons were implanted without defined tightening or angle in the axial plane.

Additionally, the study design (time-zero study) without cyclic loading does not account for the long-term effects of healing and rehabilitation on syndesmotic stability. Syndesmotic injuries often involve rupture of the deltoid ligament or fracture of the medial malleolus. These were not evaluated in this study. To reduce the number of confounding variables, we excluded medial instability [30]. The fixed sequence of the syndesmotic reconstruction may have confounded our results.

## 5. Conclusions

This biomechanical study suggests that all typical screw fixation methods, as well as double Suture Button fixation, provide greater anatomical stabilization of the tibiofibular syndesmosis than single Suture Button reconstruction, especially with regard to controlling motion in the sagittal and transverse planes. The findings also indicate that conventional a.p. radiographs may be inadequate for fully assessing syndesmotic stability and suggest that lateral radiographs could provide additional clinical information. This instability can be detected after ORIF of the ankle by a specific posterior shift of the fibula during stress testing.

## Figures and Tables

**Figure 1 bioengineering-12-00685-f001:**
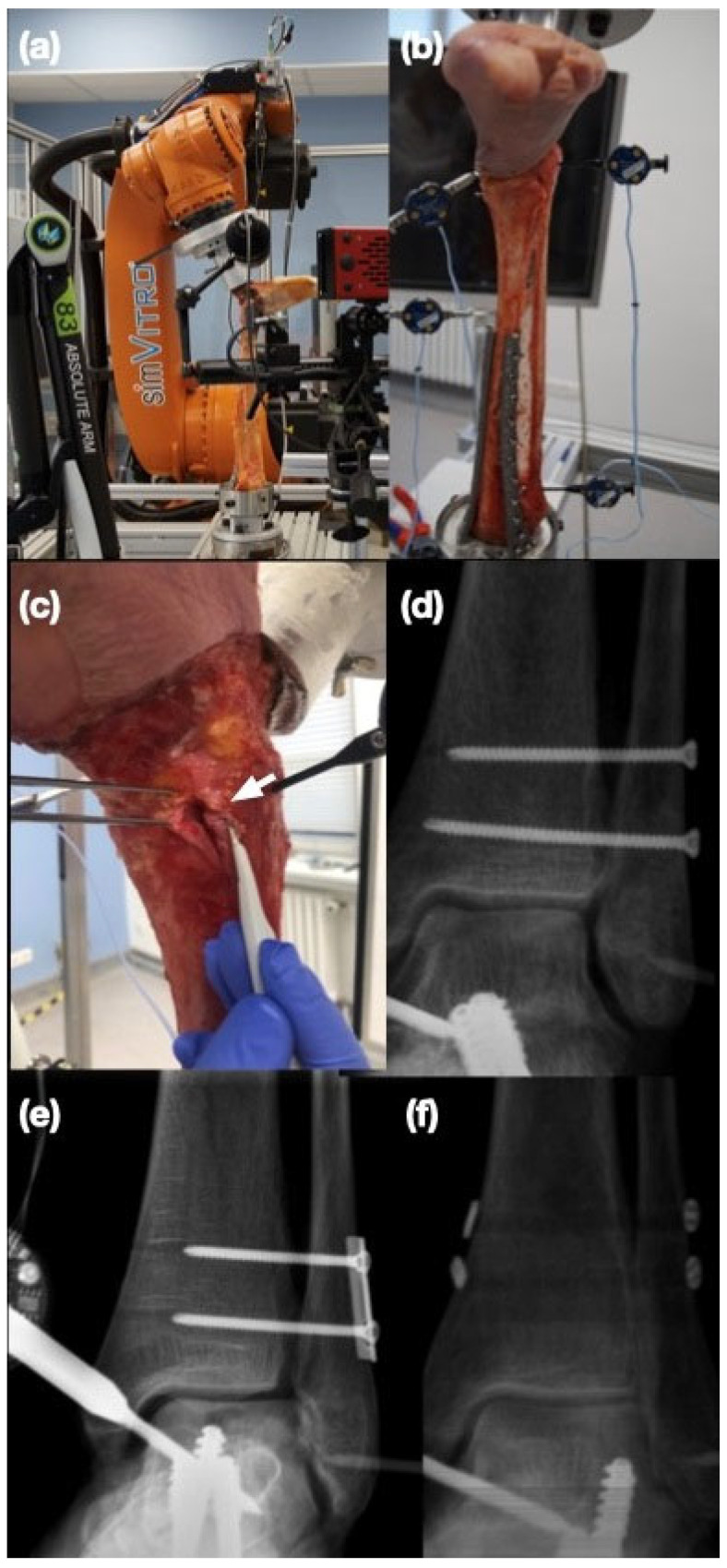
Test sequence with exemplary cadaveric left foot specimen. (**a**) Study setup with six degrees of freedom robotic arm with the specimen mounted. (**b**) Specimen mounted with optical tracking system (Optotrack Certus). (**c**) Transecting the AITFL (white arrow). (**d**) Implantation of two tricortical syndesmotic screws. (**e**) Implantation of two tricortical screws with a plate. (**f**) Implantation of two Suture Buttons.

**Figure 2 bioengineering-12-00685-f002:**
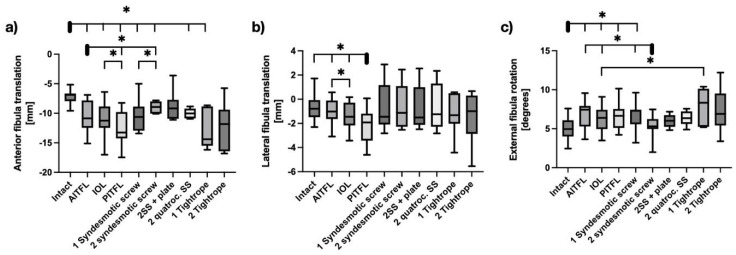
Kinematics of the distal fibula in a neutral ankle position in response to a 5 Nm external rotation stress test with sequential cutting (bars 1–4) and sequential reconstruction with screws (bars 5–8) or Suture Buttons (bars 9 + 10). (**a**) Sagittal shift. (**b**) Coronal shift. (**c**) Axial rotation. AITFL = Anterior Inferior tibiofibular ligament; IOL = Interosseus ligament; PITFL: Posterior Inferior tibiofibular ligament. Data are presented in box plots, with the median of each group in the center line, the box drawn from the first to the third quartile, and whiskers representing the minimum and maximum of each group. Horizontal lines with * represent significance, with *p* < 0.05.

**Figure 3 bioengineering-12-00685-f003:**
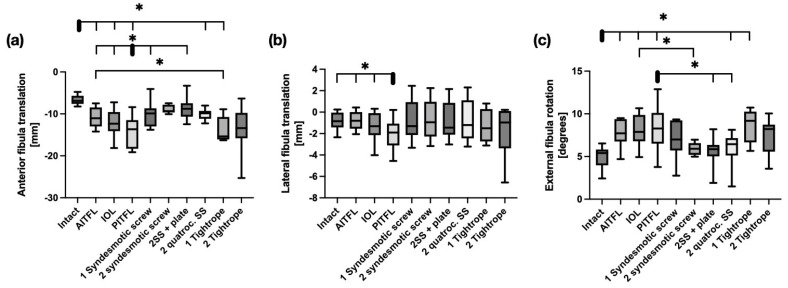
Kinematics of the distal fibula in 10° dorsiflexion in response to a 5 Nm external rotation stress test with sequential cutting (bars 1–4) and sequential reconstruction with screws (bars 5–8) or Suture Buttons (bars 9 + 10). (**a**) Sagittal shift. (**b**) Coronal shift. (**c**) Axial rotation. AITFL = Anterior Inferior tibiofibular ligament; IOL = Interosseus ligament; PITFL: Posterior Inferior tibiofibular ligament. Data are presented in box plots, with the median of each group in the center line, the box drawn from the first to the third quartile, and whiskers representing the minimum and maximum of each group. Horizontal lines with * represent significance, with *p* < 0.05.

**Figure 4 bioengineering-12-00685-f004:**
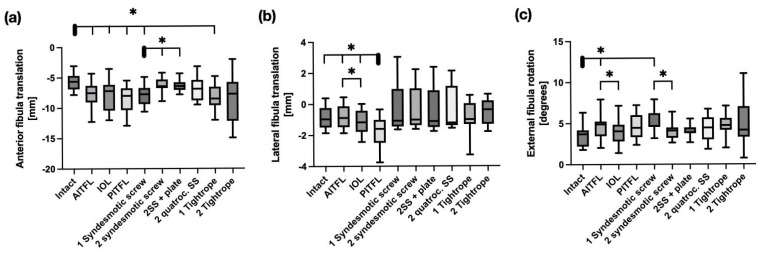
Kinematics of the distal fibula in 20° plantarflexion in response to a 5 Nm external rotation stress test with sequential cutting (bars 1–4) and sequential reconstruction with screws (bars 5–8) or Suture Buttons (bars 9 + 10). (**a**) Sagittal shift. (**b**) Coronal shift. (**c**) Axial rotation. AITFL = Anterior Inferior tibiofibular ligament; IOL = Interosseus ligament; PITFL: Posterior Inferior tibiofibular ligament. Data are presented in box plots, with the median of each group in the center line, the box drawn from the first to the third quartile, and whiskers representing the minimum and maximum of each group. Horizontal lines with * represent significance, with *p* < 0.05.

## Data Availability

The raw data supporting the conclusions of this article will be made available by the authors on request.

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
