# Peer review of "Flexible Syndesmotic Reconstruction with Two Suture Buttons Provides Equal Stability Compared to Syndesmotic Screws: A Biomechanical Study"

_bioengineering, 2025, doi:10.3390/bioengineering12070685_

Round 1

Reviewer 1 Report

Comments and Suggestions for Authors

The manuscript presents a well-structured biomechanical cadaveric study comparing different syndesmotic reconstruction techniques—specifically, traditional screw fixation and dynamic Suture Button systems.

The manuscript would benefit from a thorough English language and grammar review. A professional language edit is strongly recommended to improve readability and professionalism.

Some abbreviations are inconsistently introduced or used. For example, AITFL, IOL, and PITFL are not explained immediately when first used in the abstract or early methods. Consistent and early definition would enhance accessibility for non-specialist readers.

While the discussion acknowledges some conflicting results in the literature regarding single vs. double Suture Button fixation, it could benefit from a deeper critical analysis of these discrepancies and what specific aspects of this study design might explain the differences.

Although several limitations are acknowledged, the potential impact of implant positioning variability and absence of cyclic loading or post-operative healing simulation is not adequately addressed.

Add exact p-values to key statements in the results section, particularly when multiple reconstructions are compared.

The conclusions mention implications for radiographic evaluation. Expand briefly on how this might change current clinical protocols or diagnostic thresholds.

Author Response

The manuscript presents a well-structured biomechanical cadaveric study comparing different syndesmotic reconstruction techniques—specifically, traditional screw fixation and dynamic Suture Button systems.

We thank the reviewer for this comment.

The manuscript would benefit from a thorough English language and grammar review. A professional language edit is strongly recommended to improve readability and professionalism.

We would like to thank the reviewer for this valuable comment. We have thoroughly revised the English language and grammar with the help of native English speakers. We are pleased to conduct an additional review if necessary.

Some abbreviations are inconsistently introduced or used. For example, AITFL, IOL, and PITFL are not explained immediately when first used in the abstract or early methods. Consistent and early definition would enhance accessibility for non-specialist readers.

We thank the reviewer for this comment. The AITFL, the IOL and the PITFL are explained both in the abstract (lines 20–21), as well as in the methods section (lines 115–117).

While the discussion acknowledges some conflicting results in the literature regarding single vs. double Suture Button fixation, it could benefit from a deeper critical analysis of these discrepancies and what specific aspects of this study design might explain the differences.

We thank the reviewer for this valuable comment and have therefore enhanced the discussion about single versus double suture button fixation (see lines 278-301). However, previous biomechanical and clinical studies did not find any significant differences between the two techniques. Therefore, to our knowledge, our study is the first to demonstrate increased stability with two tightropes versus one, especially in the sagittal and axial planes. This may be due to the more precise study setup, which included optimized control of the hindfoot, precise robotic movement, and evaluation of different degrees of dorsiflexion and plantarflexion. According to the reviewer's suggestion, we also acknowledged that our statistically significant findings may not be clinically relevant since there is no consensus yet on the definition of clinically relevant rotational syndesmotic instability.

Although several limitations are acknowledged, the potential impact of implant positioning variability and absence of cyclic loading or post-operative healing simulation is not adequately addressed.

 We thank the reviewer for this comment and therefore have edited to manuscript to point out those limitations (see ll. 309-311).

Add exact p-values to key statements in the results section, particularly when multiple reconstructions are compared.

We thank the reviewer for this comment and have added p-values in the results section (see ll. 165-220).

The conclusions mention implications for radiographic evaluation. Expand briefly on how this might change current clinical protocols or diagnostic thresholds.

We thank the reviewer for this comment and have added this information to our conclusion, as the lateral radiographs might show an increased posterior shift of the fibula.

Reviewer 2 Report

Comments and Suggestions for Authors

The article submitted for review reflects a critical debate issue in modern foot and ankle trauma – the search for a fixation method for injuries of the syndesmosis. The literature on this issue is very diverse, and the proposal from the manufacturers includes information on the superiority of the method without the scientific justification. The article presents a unique and methodologically excellent example of the study, including a large number of options for the syndesmosis fixation (both rigid fixation using various options of screws and plates, and more modern, but also more controversial suture buttons).
The authors conducted a technically almost perfect experiment, including the use of robotic technology to create stress on the cadaveric preparation, as well as high-precision optical registration for measuring displacement. As a result of the experimental studies, the mechanical properties were clearly demonstrated for each of the declared methods of fixation. It was shown that the use of one suture button was of limited stability for fixation of the syndesmosis. At the same time, the use of two suture buttons gave results consistent with the use of screws. The study is nicely planned and conducted. The results are presented in aclassical sequential form. In parallel to the main conclusion about the consistency of fixation, the authors make extra conclusions about additional X-ray studies in the lateral view, which is important when planning treatment.
Among the recommendations (optional), I would like to note a more visual presentation of the material. All digital data are presented in the graphs with a rather small size and not quite the obvious figures. Perhaps it is worthwhile to accompany the article with supplementary materials reflecting a more visual representation of the experimental part, as well as numerous data at various stages of the experiment. At its core, the article answers the research question, and is recommended for publication in the present form with possible (at the decision of the authors and the editors) supplementary materials.

Author Response

The article submitted for review reflects a critical debate issue in modern foot and ankle trauma – the search for a fixation method for injuries of the syndesmosis. The literature on this issue is very diverse, and the proposal from the manufacturers includes information on the superiority of the method without the scientific justification. The article presents a unique and methodologically excellent example of the study, including a large number of options for the syndesmosis fixation (both rigid fixation using various options of screws and plates, and more modern, but also more controversial suture buttons).

The authors conducted a technically almost perfect experiment, including the use of robotic technology to create stress on the cadaveric preparation, as well as high-precision optical registration for measuring displacement. As a result of the experimental studies, the mechanical properties were clearly demonstrated for each of the declared methods of fixation. It was shown that the use of one suture button was of limited stability for fixation of the syndesmosis. At the same time, the use of two suture buttons gave results consistent with the use of screws. The study is nicely planned and conducted. The results are presented in aclassical sequential form. In parallel to the main conclusion about the consistency of fixation, the authors make extra conclusions about additional X-ray studies in the lateral view, which is important when planning treatment.

Among the recommendations (optional), I would like to note a more visual presentation of the material. All digital data are presented in the graphs with a rather small size and not quite the obvious figures. Perhaps it is worthwhile to accompany the article with supplementary materials reflecting a more visual representation of the experimental part, as well as numerous data at various stages of the experiment. At its core, the article answers the research question, and is recommended for publication in the present form with possible (at the decision of the authors and the editors) supplementary materials.

We would like to thank the reviewer for this valuable comment, and we are honoured to have received it. We have provided the raw data as supplementary files. We would like to ask the editor whether there is any need to edit the existing figures, as in our view they provide a sufficient presentation of the most important results without being too extensive and compromising the readability of the manuscript.

Round 2

Reviewer 1 Report

Comments and Suggestions for Authors

The authors have submitted a clearly improved revision of their biomechanical investigation comparing rigid (screw-based) and dynamic (Suture Button-based) fixation methods for syndesmotic injuries.

While greatly improved, several awkward or imprecise phrases remain. 

Keywords Missing.

“IOL” is listed as "Directory of open access journals," which appears to be an error. It should be “interosseous ligament.”

While the authors note that some biomechanical findings may not yet be clinically relevant, further elaboration would be beneficial. Consider stating whether a proposed clinical threshold for instability should be explored in future work.

Author Response

The authors have submitted a clearly improved revision of their biomechanical investigation comparing rigid (screw-based) and dynamic (Suture Button-based) fixation methods for syndesmotic injuries. While greatly improved, several awkward or imprecise phrases remain.

 We thank the reviewer for this comment.

Keywords Missing. 

We thank the reviewer for this comment and have added five keywords (see ll. 32-33).

“IOL” is listed as "Directory of open access journals," which appears to be an error. It should be “interosseous ligament.”

We thank the reviewer for this comment and have edited our manuscript to correct this (see l. 356).

While the authors note that some biomechanical findings may not yet be clinically relevant, further elaboration would be beneficial. Consider stating whether a proposed clinical threshold for instability should be explored in future work.

We thank the reviewer for this comment and agree completely. We have therefore edited the manuscript to emphasize the need for a defined clinical threshold for syndesmotic instability, a topic that should be addressed in future clinical studies (see ll. 305-306).